# Alkanes increase the stability of early life membrane models under extreme pressure and temperature conditions

Loreto Misuraca [1,2], Bruno Demé[2], Philippe Oger[3] & Judith Peters [1,2✉]

Terrestrial life appeared on our planet within a time window of [4.4–3.5] billion years ago. During that time, it is suggested that the first proto-cellular forms developed in the surrounding of deep-sea hydrothermal vents, oceanic crust fractures that are still present nowadays. However, these environments are characterized by extreme temperature and pressure conditions that question the early membrane compartment's capability to endure a stable structural state. Recent studies proposed an adaptive strategy employed by present-day extremophiles: the use of apolar molecules as structural membrane components in order to tune the bilayer dynamic response when needed. Here we extend this hypothesis on early life protomembrane models, using linear and branched alkanes as apolar stabilizing molecules of prebiotic relevance. The structural ordering and chain dynamics of these systems have been investigated as a function of temperature and pressure. We found that both types of alkanes studied, even the simplest linear ones, impact highly the multilamellar vesicle ordering and chain dynamics. Our data show that alkane-enriched membranes have a lower multilamellar vesicle swelling induced by the temperature increase and are significantly less affected by pressure variation as compared to alkane-free samples, suggesting a possible survival strategy for the first living forms.

[1] Univ. Grenoble Alpes, CNRS, LIPhy, Grenoble, France. [2] Institut Laue - Langevin, Grenoble, France. [3] Univ Lyon, INSA Lyon, CNRS UMR5240, Villeurbanne, France. ✉email: jpeters@ill.fr

Over the last decades there has been a growing interest in the field of the origin of life[1–4]. A number of strategies were employed to explore its key aspects, e.g., the proto-cellular compartmentalisation[5,6] or the first genetic code self-replication[7–10], sometimes resulting in contradictory conclusions. The compartmentalisation of cells, in particular, has been the focus of a large number of studies[11–16], because of the essentiality of the cellular boundary to maintain the order in living matter. These studies set the basis of the so-called "lipid world"[5,6,17], in which the formation of lipid material and the self-assembly into vesicles are considered as key phenomena that led to the appearance of life. Simple short-chain molecules with amphiphilic behaviour, e.g., fatty acids and fatty alcohols, are considered the most probable candidates as protomembrane building blocks. The rationale for this lies in the anticipated Fisher–Tropsch-driven prebiotic chemistry which likely formed such molecules[18,19], while the synthesis of phospholipid-like amphiphiles would have required a much more complex chemical pathway[20].

In terms of chemical and physical environment, there is a general consensus that life on Earth required important sources of energy and thus potentially originated under extreme conditions: some of the most accepted scenarios are the oceanic hydrothermal vents[21,22] and the terrestrial hot springs[23,24]. Both alternatives include very high temperatures (up to 100 °C) and, for deep-sea hydrothermal vents, also high hydrostatic pressures (HHP) (up to 800 bar). It is therefore crucial to include these environmental constraints when studying possible proto-membrane architectures.

The effects of high pressure on the proto-cell biochemistry have been studied extensively[25]. Several recent reviews have collected detailed information on the response of lipid assemblies to HHP (bilayer thickness increase, changes in curvature, phase transitions, interdigitation)[26,27].

The role of HHP in modifying membrane characteristics, and the possible adaptive strategies to counteract it, have been particularly investigated in the field of extremophiles[28–30]. Piezophiles, i.e. organisms which optimal growth occurs under HHP conditions, have shown the capability to tune their membrane composition in response to pressure and temperature changes of the environment[28], a process called homeoviscous adaptation. Cario et al.[28] also observed that such environmental stimuli affect

the amount of non-polar isoprenoid lipids (lycopene derivatives) synthesised by the extremophiles, suggesting a structural role of such molecules in the membrane structure. Similar apolar molecules (squalane) have been proven to promote a lipid phase separation[31] in archaeal model bilayers as well as to trigger non-lamellar phase formation at high temperature and high pressure[32]. These studies show evidences of the impact of non-polar molecules on the membrane's physico-chemical characteristics and response to extreme conditions.

However, most previous studies have been focussing on HHP effects on phospholipid or archaeal lipid membrane structure and dynamics, while little is known about the more prebiotically relevant (single-chain amphiphile based) membrane counterparts. Recently, Kapoor et al.[33] have performed pressure–temperature studies on vesicles prepared from a mixture of decanoic (capric) acid and decanol (1:1 and 2:1 molar ratio), which have much shorter chains of 10 carbon atoms length. They found that both pressure and temperature can be used to modulate the fluidity and conformational order of such protomembranes, and proved the existence of stable vesicles up to 75 °C (at 1 bar) and 2500 bar (at 25 °C).

In the present project, we performed a structural and dynamical study of multilamellar protomembrane vesicles as a function of both temperature and pressure independently, in the ranges $20 < T < 85$ °C and $1 < p < 1000$ bar. The maximum values of both variables are the conditions expected in the proximity of the hot vents. The assemblies used were decanoic acid:decanol mixtures (1:1 mol/mol), similar to the ones studied by Kapoor et al.[33], hereafter called C10 mix, which appear to be the most promising in terms of vesicle stability[34], in the presence of the apolar molecule eicosane, the linear 20-Carbon alkane (2 mol% of C10 mix), or squalane, similar length branched 30-Carbon alkane following the hypotheses of Cario et al.[28]. The differences in the membrane physico-chemical response have been measured and compared in order to gather new evidences on the effect of alkane insertion as a suitable adaptive response to extreme environmental conditions (Fig. 1).

We found that the presence of the alkanes in the membrane has a highly significant effect on the equilibrium inter-membrane spacing of the multilamellar vesicles (MLVs), a sign of its likely role in dampening membrane fluctuations at all temperatures. Squalane showed the most pronounced influence on the MLV

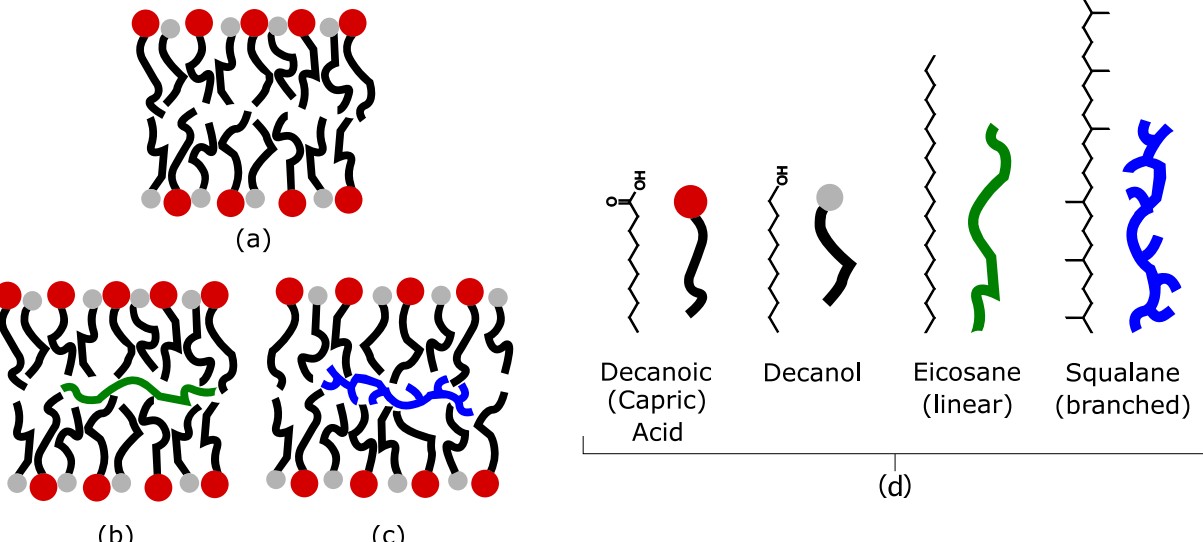

**Fig. 1 Protomembrane models.** Sketch of the three protomembrane models investigated in this study. **a** C10 mix; **b** C10 mix + 2% eicosane; and **c** C10 mix + 2% squalane. **d** Molecular structure and name of each compound used.

membrane arrangement. We observed a clear effect of both temperature and pressure in modulating the equilibrium MLV structure in the C10 mix sample, while both alkanes make the membrane less sensitive to pressure. Furthermore, the results of the dynamical study are in line with what we found for the structure modifications of the C10 mix with and without the eicosane. The average hydrogen dynamics of the membrane tails is significantly affected by temperature and pressure only when the alkane (eicosane) is not included in the sample. Upon alkane insertion, the mean hydrogen dynamics is lowered at all $p$–$T$ investigated, and it does not vary significantly even at the maximum measured values of $T = 85\,°C$ and $p = 800\,bar$.

Our results show that apolar molecules, even simple prebiotically relevant linear alkanes as eicosane, have a clear impact in modifying membrane characteristics and response to the environment. These findings shed light on a possible strategy of survival that could have helped the first living forms to cope with harsh environmental conditions while lacking the modern complex molecular tools, unavailable at the origin of life.

## Results and discussion

**Effects on membrane structuring**. We first studied the dependence of the MLV repeat distance ($d$-spacing) as a function of temperature and pressure in order to obtain information on the structural equilibrium rearrangements that are induced by the environment. Figure 2 shows an example of small-angle X-ray scattering (SAXS) curves obtained for the sample C10 mix + 2% squalane at ambient pressure (1 bar) at different temperatures. At the lowest temperature $T = 5\,°C$, two features are visible: a sharp peak centred at $q \simeq 0.17\,Å^{-1}$ (corresponding to a $d$-spacing of $\approx 37\,Å$) and a broader, swollen correlation peak centred at $q \simeq 0.09\,Å^{-1}$ (corresponding to a $d$-spacing of $\approx 70\,Å$). Details about the pressure dependence of the two coexisting phases at $T = 5\,°C$ can be found in the Supplementary Information (Supplementary Note 1 and Supplementary Figs. 1–4).

We note that very similar correlations were observed in our previous work on C10 mix MLVs (there in absence of alkanes) measured with small-angle neutron scattering (SANS) ($d \approx 37$ and $94\,Å$, respectively)[34], and where imputed to the existence of a

swollen and a collapsed lamellar phases. The latter has been interpreted as coming from a flocculated fraction of the sample which melts above the decanoic acid Krafft temperature (observed at $T \approx 10\,°C$)[34]. The Krafft temperature denotes the temperature below which the surfactant solubility (in this case, the decanoic acid) is lower than the critical micelle concentration, so that no micelles or vesicles can form anymore and flocculation occurs. Therefore, at $T = 5\,°C$ we are in the presence of a two-phase system: one is the MLV phase, with weakly interacting bilayers that leads to the broad correlation observed at lower $q$ (swollen phase); the other one, which can be thought as made of microscopically phase separated aggregates, that gives a sharp correlation in the scattering curve at higher $q$ (collapsed phase).

The curve at $T = 20\,°C$ (Fig. 2) captures an intermediate state where the melting of the collapsed phase is almost completed. At $T = 35\,°C$, one main correlation is observed at $q \simeq 0.08\,Å^{-1}$ (the second order can be detected at $q \simeq 0.16\,Å^{-1}$ although weak). At $T = 50\,°C$, the correlation is further shifted to lower $q$ and broadened, hardly detectable by visual inspection. At $T = 65\,°C$ one can only guess the position from the trend at low temperature, and finally at $T = 80\,°C$ the correlation is completely lost. This behaviour, with the position of the correlation shifting to lower $q$ until disappearing, is expected for MLVs that undergo swelling upon temperature increasing, until unbinding[35,36]. In our analysis, we calculated the $d$-spacing of the three samples at $T \leq 50\,°C$ only.

The data from the C10 mix sample at $T = 20\,°C$ and $p = 10\,bar$ show a swollen, broad phase centred at $q \simeq 0.05\,Å^{-1}$ ($d$-spacing $\approx 122\,Å$) together with a small, thin correlation at $q \simeq 0.08\,Å^{-1}$ ($d$-spacing $\approx 75\,Å$) (Supplementary Fig. 4). The details and the interpretation of such coexistence, observed at all temperatures for this sample, can be found in the Supplementary Information (Supplementary Note 1). In the following, we focussed our investigation on the correlations related to the most swollen phases, characteristics of the fluid and weekly interacting membranes.

All curves for each $p$–$T$ point were fitted in order to find the average $d$-spacings, as discussed in the "Methods" section. The results are plotted in Fig. 3 for the three samples measured. The sample containing squalane at 20 °C is missing, because the still ongoing melting of the collapsed phase did not allow a reliable fit of the swollen phase correlation (as it can be inferred from Fig. 2).

A number of insights can be drawn from these results. First, there is a clear effect of the alkane presence in the equilibrium MLV $d$-spacing as a function of temperature. Both alkanes lower the $d$-spacing significantly, with the squalane leading to the most prominent shift. Second, there is also a clear effect of pressure on the C10 mix sample in lowering the $d$-spacing. This is likely to be a sign of the membrane fluid–gel transition, which has been observed to occur at 10 °C at ambient pressure[33,34]. Here the transition seems to happen from $p \simeq 300\,bar$ at $T = 20\,°C$ in agreement with Fourier-transform infrared (FTIR) measurements (Supplementary Note 3 and Supplementary Fig. 6). Assuming the linearity of the fluid–gel phase transition with $p$–$T$ variation, this leads to a shift of $\approx 3\,°C/100\,bar$. This value is similar with what has been observed with phospholipid membranes ($\approx 2\,°C/100\,bar$)[37]. Following this relationship, the phase transition of the C10 mix at $T = 35\,°C$ is expected at $p \approx 800\,bar$, although our data do not allow to conclude unambiguously. An analogous pressure effect is much less evident on the C10 mix with eicosane (only a possible shift might occur at $p \simeq 800\,bar$ at $T = 20\,°C$) and squalane, where the $d$-spacing is instead increasing slightly with pressure. Indeed, the synergistic effect of pressure and temperature can lead to both kinds of variations in $d$-spacing in relation with phase transitions, as previously shown for the phospholipid DMPC[38].

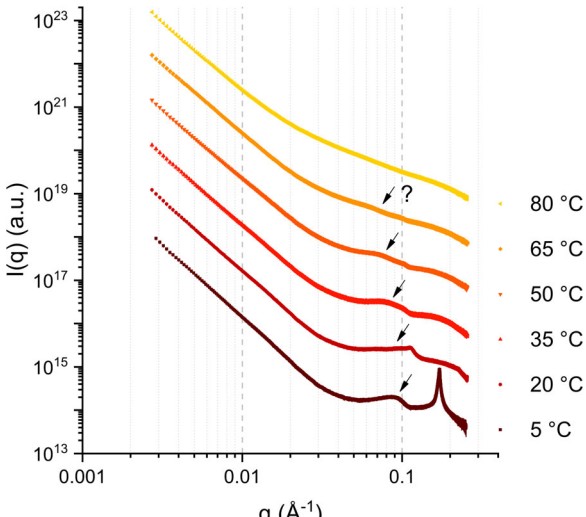

**Fig. 2 Example of SAXS curves.** SAXS curves obtained for the sample C10 mix + 2% squalane at $p = 1\,bar$. The errors are calculated by propagating the errors of the 30 averaged frames (which come from Poisson distribution). Most of the error bars are smaller than the symbol size. All curves were vertically shifted for clarity.

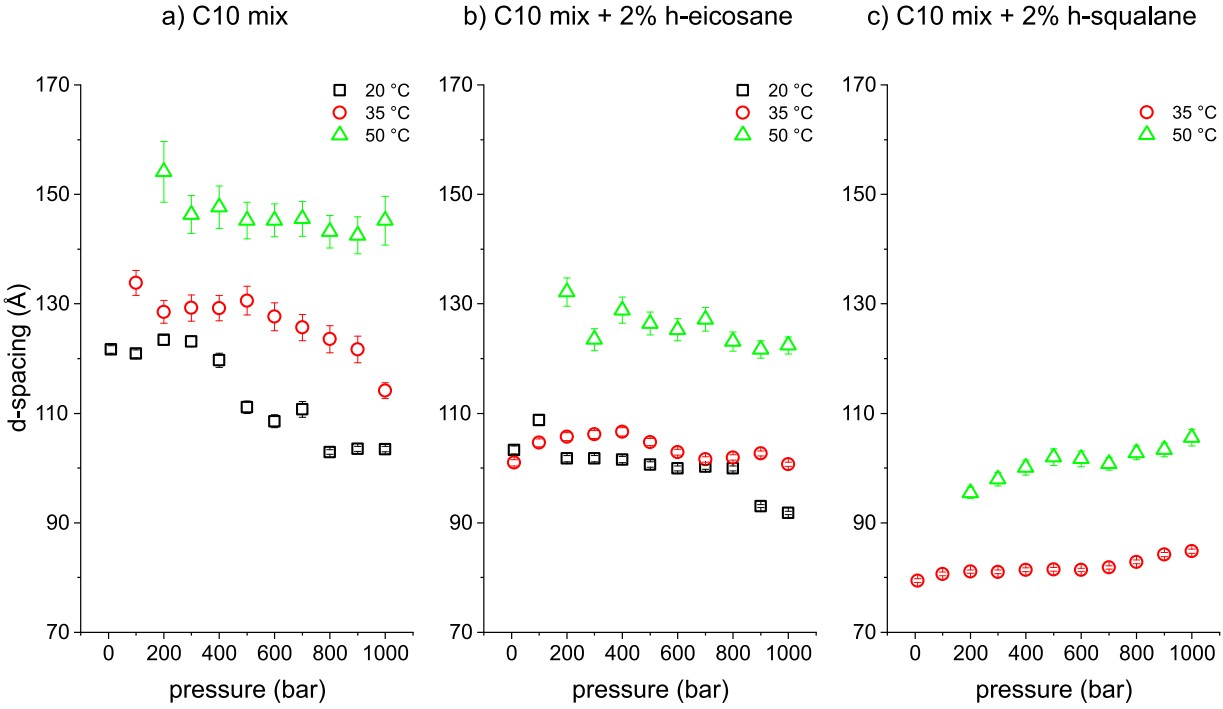

**Fig. 3 d-spacing vs p–T.** MLV d-spacing of the three measured samples at all T–p points where a lamellar correlation was fitted. **a** C10 mix; **b** C10 mix + 2% h-eicosane; and **c** C10 mix + 2% h-squalane. The errors are calculated by propagation of the fit parameter errors, as detailed in the main text.

**Effects on membrane dynamics**. The difference between samples missing or containing the alkanes was further investigated by studying the mean hydrogen dynamics of the amphiphile chains forming the membranes. The decanoic acid and decanol hydrophilic groups (R-COOH and R-OH, respectively), being partially ionised at the buffered pH conditions, will contribute with very few hydrogens and therefore we can neglect their contribution.

Elastic incoherent neutron scattering (EINS) experiments were performed on C10 mix with and without perdeuterated eicosane, as a function of temperature and pressure. The use of perdeuterated eicosane allowed ensuring that most of the incoherent scattering signal was coming from the amphiphiles in the membrane, the main object of our study.

The calculated mean square displacement (MSD) values (see the "Methods" section) are plotted in Fig. 4 as a function of the temperature and at the different pressure points. All MSD present very high values what is typical for lipidic motions[39].

The data show clear differences between the MSD of C10 mix with and without eicosane at almost all pressure and temperature values applied. The two samples start from similar MSD values at 25 °C and all pressure points. At the higher temperatures, eicosane keeps the hydrogen MSD to a lower value close to the one at T = 25 °C: this can be interpreted as the membrane maintaining a higher rigidity compared to the pure C10 mix one. Conversely, the C10 mix lacking the eicosane shows a more pronounced increase in dynamics upon temperature increase, with an effect that is inversely proportional to the applied pressure.

These results are in line with what has been shown in our SAXS data on C10 mix and C10 mix + 2% eicosane samples (Fig. 3). In fact, at high d-spacing, the dominating contribution responsible for the water layer thickness is mostly given by the membrane thermal fluctuations, while all short-range interactions are negligible at such membrane separations. A stiffer membrane, maintaining a lower MSD value as observed for the sample containing eicosane in Fig. 4, will therefore lead to a shrunken

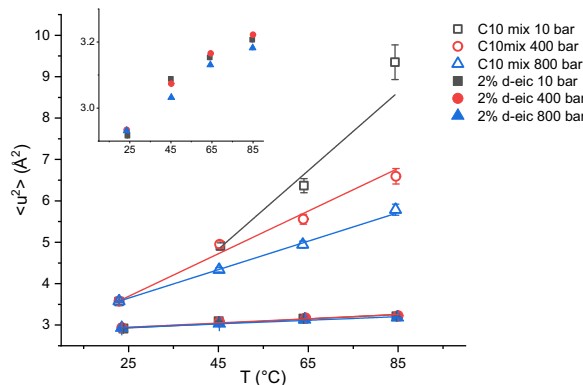

**Fig. 4 Atomic mean square displacements.** MSD for the two samples studied: C10 mix (empty symbols) and C10 mix with d-eicosane (full symbols). All lines are linear fits to the data. Note the clear dependence on temperature and pressure for the sample missing the eicosane, while all MSD values vary little at all T–p when the eicosane is added. The errors are calculated by propagation from $I_{sum}$, as detailed in the main text. Inset: vertical zoom of the C10 mix + 2% d-eicosane data.

MLV and a smaller d-spacing (which is what is observed, see Fig. 3a, b).

Additional insights can be obtained by performing a simple linear fit to the MSD vs T data (solid lines shown in Fig. 4), in order to obtain qualitative information about the pseudo-force constant characteristic of the particular sample and pressure employed, in a similar fashion as described by Zaccai[40] for protein dynamics. In this representation, the slope of the line, namely $du^2/dT$, gives an estimation for the pseudo-force constant to be compared among the different samples and pressure points. Note that, as the data of C10 mix at p = 10 bar show an important but not linear increase in the dynamics, we performed a linear fit

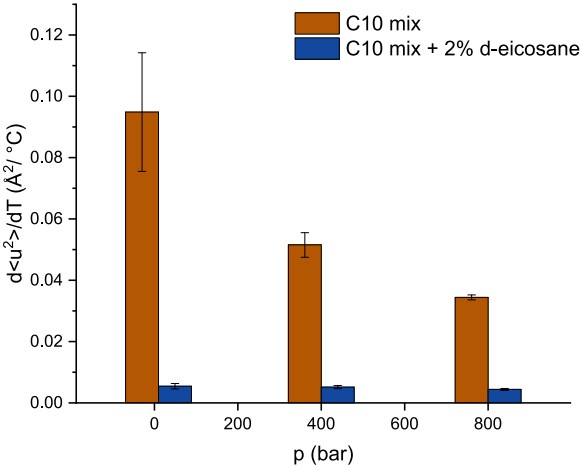

**Fig. 5 Pseudo-force constants.** Histogram comparing the derivative of the MSD data vs temperature assuming a simple linear model. Note that the value of C10 mix at $p = 10$ bar is a likely underestimation. Each value is shown with its error from the linear fit.

without including the $T = 25\,°C$. Even with this strategy, the strong increase in the dynamics at $T = 85\,°C$ is higher than what is expected for a linear behaviour; therefore, the corresponding pseudo-force constant value must be considered as an underestimation. All the slopes are compared and depicted in Fig. 5.

This representation allows for a clear differentiation between the two systems: without eicosane, the $du^2/dT$ is significantly affected by the applied pressure. Instead, the addition of the alkane causes the membrane to become almost pressure insensitive. This is in full agreement with what has been found from the structural study by SAXS on the sample with and without the eicosane (Fig. 3a, b). Note that the derivative $du^2/dT$ is positive also on the eicosane including sample, which means that the temperature still causes an increase in the MSD in this sample, although of much lower extent as compared to the C10 mix (Fig. 5).

Taken together, these results show that the insertion of alkanes inside model protomembranes, composed of short-chain amphiphiles, has an important impact on the membrane structuring and dynamics. In particular, both alkanes investigated in the structural study (eicosane and squalane) lower the equilibrium $d$-spacing of the MLVs and cancel the effects that are due to the increase of hydrostatic pressure. Squalane leads to the biggest effect on the structural MLV characterisation, which could explain the findings of branched alkanes being used by modern Archaea[28] as one of the adaptive strategies when facing high-pressure environments. Moreover, the dynamical study proved that the eicosane decreases significantly the effects of HHP and high temperature on the motions of the acyl chains in the membrane.

The observed phenomenon, mediated by a small proportion of alkane molecules, could help explaining how the first living forms have survived the harsh thermodynamic constraints imposed when considering one of the most currently accepted scenarios for the origin of life (i.e. deep-sea hydrothermal vents). Such strategy, for pressure enduring and desensitisation, could also be used to readily explain how those systems could have afterwards migrated from high-pressure environments towards ambient pressure ones, leading to the biosphere as we know it nowadays.

## Methods

**Sample preparation**. Sodium decanoate, 1-decanol, eicosane (hydrogenated and perdeuterated), squalane, bicine buffer and $D_2O$ were purchased from Sigma Aldrich (Merck). All products were used as received from the manufacturer, with

no further purification. The chemical and isotopic purity (CP–IP) of the used compounds are the following:

sodium decanoate: ≥98% CP;
1-decanol: ≥98% CP;
h-eicosane: 99% CP;
d-eicosane: 98% CP; 98% IP;
h-squalane: 96% CP;
d-squalane: 98% CP; 98% IP;
bicine: ≥99% CP.

The samples were prepared by first mixing the decanoic acid and the decanol (plus the eicosane/squalane, when needed) in bulk at the proper amount to obtain a 1:1 decanoic acid–decanol final molar ratio (plus 2% molar of the corresponding alkane). Samples were then dissolved in a $CHCl_3$ (Merck) solution to ensure complete mixing, followed by drying under a flux of nitrogen until no evaporation was observed. All samples were placed in a desiccator and left under vacuum overnight. The samples, checked gravimetrically at each step of the preparation, showed a loss of ≈5% that was imputed to a partial evaporation of the decanol[34].

The bicine buffer was prepared at a concentration of 0.2 M in $H_2O$ for standard experiments, and in $D_2O$ for the neutron scattering experiments. The $D_2O$ buffer allowed to minimise the background due to the hydrogen incoherent neutron scattering. In all cases, the buffer was filtered with a 0.2-μm millipore membrane before use and titrated to pH/pD 8.5 with aliquots of NaOH/NaOD. When adjusting the pD of the $D_2O$ solutions to 8.5, we used the formula pD = pH* + 0.4 (ref. [41]), where pH* is the value measured by an $H_2O$-calibrated pH meter.

Previous work on decanoic acid:decanol systems[33] has shown that, unlike the vesicles made by decanoic acid that are heavily dependent on the solution pH, the ones consisting of the decanoic acid:decanol 1:1 mixture (which we named C10 mix) are stable in a wide pH range (pH 6–12). Furthermore, the temperature-dependent change in the $pK_a$ of bicine buffer is rather low ($dpK_a/dT = -0.018$)[42]. We have chosen pH (or pD) 8.5 so that the minor changes in the buffer $pK_a$ and solution pH given by the temperature increase do not affect the vesicle stability.

The dried organic solutions were suspended in the corresponding buffer for the neutron and X-ray experiments and vigorously vortexed for ≈1 min, leading to final milky solutions. The sample concentration for the X-ray scattering experiments was set to 350 mM, already employed in a previous work[34] on C10 mix MLVs. For neutron scattering experiments, the concentration used was 100 mg/ml corresponding to ≈570 mM in order to obtain a sufficient signal-to-noise ratio. For the complementary neutron scattering experiments, the samples were prepared at 80 mM concentration. The three model membranes measured were:

1. C10 mix
2. C10 mix + 2% (h/d) eicosane
3. C10 mix + 2% (h/d) squalane.

All samples were measured once at each $p$–$T$ point.

**Small-angle neutron scattering**. SANS experiments were carried out at the ILL (Grenoble, France) using the D33 instrument[43]. Two detector distances (2 and 10 m) and an incoming beam of $\lambda = 5$ Å were used, corresponding to a range of momentum transfers $0.004 < q < 0.5$ Å$^{-1}$. The sample suspensions, of 200 μL each, were loaded in 1 mm quartz cells (Hellma, Germany) on a sample holder with thermal regulation. All data were corrected for the scattering of the sample container and the instrumental background. $H_2O$ was measured for detector efficiency calculation and scaling to absolute units (cm$^{-1}$). A flat background was subtracted to account for the $q$-independent incoherent neutron scattering signal.

**Small-angle X-ray scattering**. SAXS experiments were performed on I22 at the Diamond Light Source (Didcot, UK)[44].

An automated high-pressure cell, adapted for SAXS experiments and available as sample environment on the I22 beamline, was used[45]. The samples were loaded into capillaries in 50 μl aliquots and sealed using a glued rubber cap. An empty capillary measurement was performed for background estimation and subtraction. For each acquisition, 30 frames of 2 s were collected and averaged afterwards to avoid saturation of the PILATUS SAXS detector. Every sample was measured at 5 < $T$ < 80 °C (with steps of 15 °C) and 1 < $p$ < 1000 bar (with steps of 100 bar), for a total of 66 $p$–$T$ points. The scans were performed as isotherms, setting the desired temperature, waiting 600 s for sample equilibration and measuring the first pressure point. The sample was then brought to the successive pressure step, and a 5 s equilibration time was set before the new acquisition. The same was repeated for all $p$–$T$ points.

The I22 beamline at Diamond Light Source (Didcot, UK) is equipped with a fast shutter that is used to make sure that the samples are only illuminated during the short counting times and not during $T$–$p$ changes and equilibration time. The 17 keV X-rays used for the high-pressure experiments on I22 helps reduce the radiation damage to biological samples[45]. If radiation damage occurred to the samples, this would have resulted in a decrease of the SAXS intensity vs time since the $T$–$p$ scans were performed on the same sample for a given composition. No decrease in intensity was observed.

The resulting images were radially averaged, leading to a series of one-dimensional intensity $I(q)$ vs $q$ curves, where $q$ is the module of the momentum

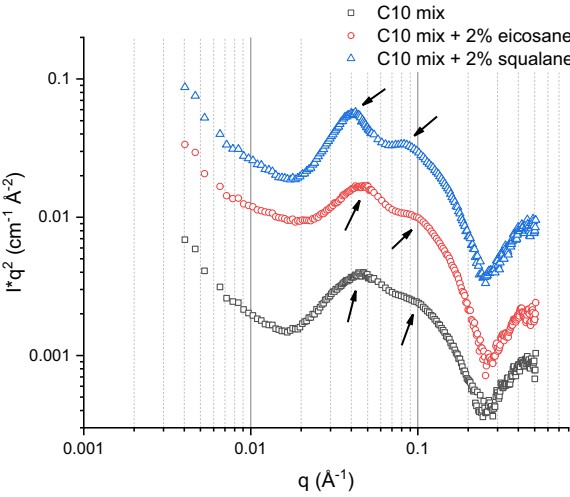

**Fig. 6 Sample lamellarity at ambient temperature.** SANS curves obtained for the three samples at 80 mM concentration and $T = 21\,^\circ\text{C}$. The arrows indicate the position of the first two orders of the MLV correlation for the three samples. The curves of eicosane- and squalane-enriched samples were shifted vertically for clarity.

transfer vector and is linked to the scattering angle by $q = 4\pi \sin\theta/\lambda$ ($\theta$ is the scattering angle and $\lambda$ the incident wavelength). The intensity $I(q)$ consists of two terms:

$$I(q) = \left| F(q)^2 * S(q) \right| = \left| P(q) * S(q) \right| \qquad (1)$$

where $P(q) = F(q)^2$ is the particle (in our case, the vesicle) form factor and $S(q)$ here is the inter-membrane structure factor which is of particular interest in this study. It contains the information about the interaction and the ordering of the membranes within the MLVs. By localising the $q$ positions (which are a feature in the reciprocal space) of the first correlation maxima, and converting them to the real space membrane periodicity via the relation $d_{\text{spacing}} = 2\pi/q$, one obtains the mean membrane $d$-spacing. This quantity is the average value of the membrane repeat distance within MLVs, i.e., sum of the mean bilayer thickness and the mean buffer layer thickness between two adjacent membranes. We verified that the observed correlation is due to a lamellar phase of MLVs from the position of the second-order correlation peak with respect to the first order: $q_{\text{2nd}} = 2 \times q_{\text{1st}}$ at ambient pressure (Fig. 6) as well as at high pressure (Fig. 7).

Because of the weak $S(q)$ observed and as the main focus was the MLV $d$-spacing in this study, we did not perform a full fitting of the entire SAXS curves. Instead, we exploited the possibility for the vesicle form factor $P(q)$ to be approximated by a power law decay in the $q$-range where the first MLV correlation maxima are observed[34,46]. Therefore, the location of each correlation maximum was extracted by fitting Gaussian functions in the range of interest, with a background of the form $q^{-k}$ (with $k$ left as a free parameter for each specific temperature and sample, shared for all pressure points) to eliminate the contribution from the $P(q)$. Such a $q^{-k}$ trend is observed as a linear background in log–log plots. The parameters corresponding to the centre of the Gaussian functions where converted into $d$-spacing values (via $d_{\text{spacing}} = 2\pi/q$), as well as the associated errors (through error propagation). Figure 7 shows an example of fits performed.

**Elastic incoherent neutron scattering**. The incoherent neutron scattering experiments were performed on the backscattering spectrometer IN13 at the Institut Laue-Langevin (ILL Grenoble, France)[47]. Using backscattering geometry, one can access a very high energy resolution ($\Delta E \approx 8\,\mu\text{eV}$) which translates into an observable timescale of motions of $\approx 0.1$ ns. In such experiments, the signal obtained is dominated by the incoherent scattering coming from the hydrogen nuclei. In the case of the samples studied here, it means that the signal arises mostly from the acyl chains in the membrane, because the solvent used (and also the alkane molecules) were perdeuterated. This allowed exploiting the large difference in the scattering cross section of the two hydrogen isotopes to only highlight the acyl chain dynamics. The output of such an experiment is a series of curves $I_{\text{inc}}(q, E)$ vs $q$, where the rate of intensity decay gives quantitative information on atomic MSDs at every $T$–$p$ point investigated.

The HHP equipment for biological samples in solution was developed in collaboration with the Sample Environment group (SANE) of the ILL. It consists of a pressure controller, which communicates with the instrument control software NOMAD, a high-pressure stick[48] and an HHP cell[49]. The pressure is transmitted from the controller to the sample through a capillary using liquid Fluorinert™[50], which has a pour point of 178 K. As the stick is inserted in the cryostat to regulate

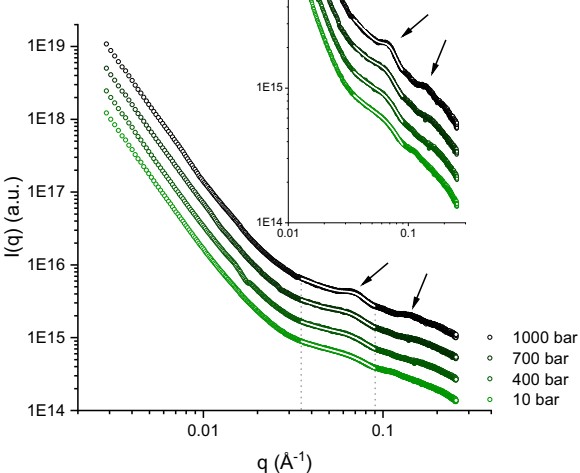

**Fig. 7 Fitting the SAXS data.** Example of SAXS data fitting using Gaussian functions with a $q^{-c}$ background. The curves show the SAXS signal from the sample C10 mix + 2% eicosane, $T = 20\,^\circ\text{C}$ and $p = 10, 400, 700, 1000$ bar respectively. Arrows show the two first correlation orders, proving the membrane lamellar ordering (together with the data shown in Fig. 6). The white lines are fits to the data. Inset: zoom in the mid-wide $q$-range. All related fitting parameters, errors and resulting $\chi^2$ can be found in the Supplementary Information (Supplementary Table 1, see also Supplementary Note 2 and Supplementary Fig. 5).

temperature, one has to avoid that liquid freezes; therefore, it must be heated by a wire and isolated thermally from its environment by a secondary vacuum. The HHP cell, made of the high-tensile aluminium alloy 7075-T6, is cylindrical with an external diameter of 10 mm and an internal diameter of 6 mm. It withstands pressures up to 1 kbar. The sample solution was separated from Fluorinert™ by a separator on the top of the cell.

Given the spectrometer resolution, the signal is generated by specific molecular motions within the corresponding timescale: methyl rotation, amphiphile rotation along its main axis and in-plane diffusion within the membrane[51]. Every sample was measured at four temperatures (25, 45, 65 and 85 °C) and three pressure values (10, 400 and 800 bar), for a total of 12 $p$–$T$ points. For every measurement, 2 ml of sample were loaded in the HHP cell. The empty cell and a Vanadium rod measurement were performed in order to subtract the background and to normalise the data to the signal of a totally elastic incoherent scatterer.

Each $I_{\text{inc}}$ vs $q$ curve obtained per $p$–$T$ point was analysed using a model-free approach described in ref. [52]. In the limits of the Gaussian approximation (GA)[53], which assumes harmonic motions of the atoms around their equilibrium positions and thus a linear behaviour of $\log(I_{\text{inc}})$ vs $q^2$, one can write the sum of the intensity curves over $q$, namely $I_{\text{sum}}(T) = \sum_{q_{\text{min}}}^{q_{\text{max}}} I_{\text{inc}}(q, T)$, as follows:

$$I_{\text{sum}}^2(T) \propto \frac{1}{\langle u^2 \rangle} \text{Å}^2 \qquad (2)$$

with $\langle u^2 \rangle$ the hydrogen MSD and Å the angstrom unit. This gives a direct way of measuring the MSD and the corresponding error by simple error propagation from $I_{\text{sum}}$ and allows to profit from a better statistics. The errors on $I_{\text{sum}}$ are only dependent on the Poisson distribution of the scattered neutrons. The MSD, a measure of the mean hydrogen dynamics, can therefore be compared between the different samples.

The data range to be considered, when working in the GA, should satisfy the validity condition $\langle u^2 \rangle q^2 \leq 1$ (ref. [54]). In our case, from the reasonable assumption that all our samples consist mainly of spherically shaped objects (the MLVs), one can consider a range slightly beyond the validity condition[54]. Thus, the final range was defined by finding the maximum $q$ for which the curve of highest dynamic (C10 mix $T = 85\,^\circ\text{C}$, $p = 1$ bar) shows a linear trend in the $\log(I_{\text{inc}}(q))$ vs $q^2$ representation. Some examples of $I_{\text{inc}}$ curves and the corresponding range of validity of the GA are shown in Fig. 8.

A normalisation factor, common to all $p$–$T$ points and samples measured, was applied on the MSD values to obtain results quantitatively comparable with the ones found from the standard GA fits (via the formula $I_{\text{inc}}(q) \cong I_0 \exp[-\langle u^2 \rangle q^2/3]$[53]). The latter fits lead to absolute $\langle u^2 \rangle$ values (and thus can be used for normalisation) but they are highly affected by the fitting errors, justifying our use of the alternative fit- and model-free method.

In principle, additional information could be inferred by extending the analysis to the high $q$-range, using a more complex model[55] or by considering two dynamic populations (low-$q$ region–high-$q$ region) instead of a single one[52]. Nevertheless, we focussed only on the largest amplitude motions (encoded in the lowest $q$-range)

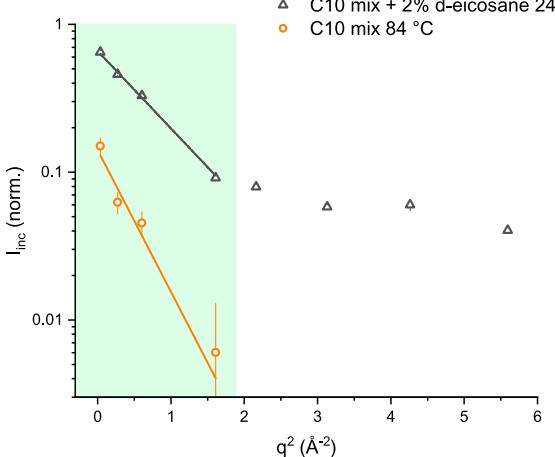

**Fig. 8 Elastic intensity decay and GA validity range.** Example of intensity curves obtained from the EINS experiment and range of validity of the GA (shown as a linear behaviour in this representation). The data shown are the ones of C10 mix and C10 mix + 2% eicosane at $p = 10$ bar and $T = 84$ and 24 °C, respectively. The light green region highlights the $q^2$ data range used for the analysis. The $q^2 > 2$ Å$^{-2}$ data for the C10 mix at $T = 84$ °C approach zero and are not visible in the $\log(I_{inc})$ representation. The errors are obtained from Poisson distribution.

because this is the region specifically related to the chain motions[56] and given the very high dynamics of the C10 mix sample which causes a drop of $I_{inc}$ towards zero for $q^2 > 1.6$ Å$^{-2}$ (also visible from Fig. 8).

## Data availability

The SANS data that support the findings of this study are available with the identifier Doi: 10.5291/ILL-DATA.9-13-905. The SAXS data are available from the corresponding author upon reasonable request. The EINS data are available with the identifier Doi: 10.5291/ILL-DATA.9-13-828. The FTIR data (Supplementary Fig. 6) are available from the corresponding author upon reasonable request.

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

## Acknowledgements
This work was supported by the French National Research Agency programme ANR 17-CE11-0012–01 to P.O. and J.P. J.P. was funded by Campus France, programme Hubert Cuprien, with the German-French bilateral research cooperation programme "Procope" 2018–2019 (Contract NR 39974VD). L.M. was supported by a scholarship from the Institut Laue-Langevin (ILL) PhD program. The authors thank ILL for neutron beam-time on IN13. The work was carried out with the support of Diamond Light Source (Didcot, UK), instrument I22 (proposal SM23722). We acknowledge Olga Shebanova for her support during the experiment on I22. The ILL Partnership for Soft Condensed Matter (PSCM) is acknowledged for the access to the lab infrastructures. This work benefited from SasView software, originally developed by the DANSE project under NSF award DMR-0520547 [http://www.sasview.org/]. We are grateful to Josephine LoRicco for her support during the experiments. We gratefully acknowledge Antonio Calio for his help during the EINS data re-analysis. We thank Roland Winter for his help with the FTIR measurements.

## Author contributions
This study is part of the Ph.D. project of L.M., carried out under the joint supervision of B.D., P.O. and J.P. L.M. conceived and designed the experiments, analysed the data and wrote the paper. L.M., B.D. and P.O. performed the SAXS experiments. L.M., P.O. and J.P. performed the EINS experiment. L.M., B.D. and J.P. performed the SANS experiment. B.D., P.O. and J.P. jointly supervised the data analysis and contributed to the paper redaction.

## Competing interests
The authors declare no competing interests.
