## [Peer Review File · Communications Chemistry]

Reviewers' comments:

Reviewer #1 (Remarks to the Author):

The authors show data that demonstrate the reduced mobility of a model bilayer system composed of capric acid and decanol, that is aimed to resemble the first bilayer structures formed in the process for life formation. These two molecules experience increased movement upon increasing temperature, and this is counteracted by pressure. Life is thought to have formed in regions of high temperature and pressure, therefore the authors study the effect of bilayer structure (in terms of d-spacing) and bilayer motions (general motions) upon incorporating two alkanes: a linear one and a branched one. By using SAXS, the authors follow the change in d-spacing and observe how the alkanes decrease this value at all temperatures and pressures. By using EINS and perdeuterated alkane, the collective motion of the acyl chain in capric acid and decanol were shown to be maintained at all temperatures and pressures to values similar to those at 25°C. This was in contrast with the samples measured without added alkane, that show a continuous increase in motion with temperature at all pressures. Together the data show that the presence of the alkanes reduces the bilayer mobility (and increase bilayer rigidity) enabling the bilayer structure to be retained even under very harsh conditions. The work is very convincing and well presented. I wonder what the authors think is the structure of these bilayers? Where are the added alkanes positioning? Will they be freely mixed in the bilayer structure or is there a preference to occupy a certain position within the bilayer?

There are still some questions I hope to be addressed prior to submission:

- state the number of samples measured.
- state how the error bars in Figure 5 and 6 are deduced from.
- why are the error bars in Figure 6 for data in absence of added alkane so much larger than for the data with added alkane?
- what is the purity of the perdeuterated eicosane?
- the condensed phase formed? is it thought as two different vesicles populations, one that falls out of solution and another one that remains in solution (the fluid phase)?

Reviewer #2 (Remarks to the Author):

The paper submitted by Misuraca et al. is dealing with the structure and dynamics of protomembrane model systems in order to get some insight into early strategies by nature to adopt functional membranes surviving the harsh environment of early life. The authors have applied small angle x-ray scattering (SAXS) and elastic incoherent neutron scattering (EINS) to probe structure and dynamics of simple model systems of mixed fatty acid / fatty alcohol systems in the presence of certain alkanes at various temperature and pressure values. The found effects on the structure and the local dynamics of the alkyl chains in these model systems indicate, argued by the authors, a stabilizing effect of the added alkanes on the membrane resistance towards temperature and pressure changes. This behavior is seen as a possible strategy of survival for the initial forms of life.

The paper is reasonable well organized and the description of the experiments conducted is quite sound. In order to consider a publication of the work presented nevertheless a number of improvements should be taken into account, as the presentation and discussion of some of the results should be improved critically. In the following these points will be discussed.

- In the introductory part the authors describe the reduction in intermembrane space as "a sign of its (alkanes) role in dampening membrane fluctuation". Usually large intermembrane spacings are an effect of electrostatic repulsion and are triggered mainly by changes in ionic strength or pH. The large d-spacing observed and related to a so called "swollen phase" may indicate a weak

correlation between individual membranes which might be a transient interaction averaged over the time of the sampling of the SAXS experiment. The shown changes in d-spacing (Fig. 5) may reflect this transient correlation and the found decrease by presence of alkanes could be interpreted as a damping of membrane fluctuations or more general of the dynamic of the membrane fluctuations.

On the other hand the temperature dependence of this interaction seems to get lost in the presence of the alkanes as on changing the temperature from 20 to 35 oC has no substantial effect anymore on the d-spacing (Fig.5, middle). It is not clear why this should happen and the simple assumption of the damping of membrane fluctuations seems to be more complex. There is also no substantial effect in the MSD shown in Fig. 6 on increasing the temperature in the systems with eicosane, which should exhibit an increase. Here, the authors should revise their interpretations and be more careful in the description of the drawn conclusion on this behavior.

- The authors should make a comment regarding beam damage onto the samples by the exposure to the synchrotron radiation.

- The authors have not tried to do a full fitting of the collected SAXS curves (see p.4, line 153). With relation to the assumed changes in the interaction between the individual bilayers in the MLV it would be interesting if the sizes of the MLV can be fitted with a classical sphere or ellipsoidal model, which would indicate a shape fluctuation. It would also be interesting to have seen some electron microscopy pictures to prove the multilamellar arrangement proposed.

- On p.7, line 222 the authors should write "a drop of I_{line} towards zero..." as zero is not reached.

- The description of the collapsed phase as a flocculated fraction of the systems is difficult to follow (p.7, line 236 ff). Do the authors mean a two phase system or a phase separation? This should be explained more clearly.

- The authors mention a "a weak undulation ... in the q range 0.1-0.2 Ang⁻¹" (p.8, line 248). This is in no way visible in the presented graphs. The authors should give more clear evidence on this assumption.

- On p.8, line 250 ff the q and d-spacing numbers are missing in the text.

- The authors state the fluid-gel transition at 35 oC to be at 1000 bar. Following Fig. 5 this transition is at about 500 bar. In Fig. 5 the authors should not connect the points measured by arbitrary curves but with straight lines to visualize the phase changes and error ranges.

- The results of the EINS data shown in Fig. 6 imply a linear change in MSD for the mixed system without alkanes for different pressure applied. A more closer look at the data shows an increase between 25 to 45 oC and then a roughly linear change or even decrease within the error range given towards higher temperatures. It looks as if the simple conclusion of a linear increase in MSD is not fully valid. The authors should carefully reanalyze this data as it seems to resample partly the different behavior in d-spacing with temperature shown in the SAXS data in Fig.4 and changed due to the fluid-gel transition at applied pressure.

- On page 11, line 327 ff the authors claim squalene has the biggest effect on the structure and dynamics investigated. Unfortunately, the authors have not shown any results on the EINS with squalene. It is assumed squalene will have a more clear or similar effect as found with eicosane, but this is only an assumption derived on the basis of the SAXS data. The authors should be slightly more cautious on this conclusion.

- A minor point is the use of the term vesicle. To avoid confusion and to describe the work presented correctly the authors should always use the term "multilamellar vesicles" in the manuscript as only multilamellar systems have been investigated. At increased temperature the

multilamellar vesicles obviously transform into unilamellar vesicles but there is no experimental proof given throughout the manuscript on this assumption, therefore it should be avoided to describe the systems investigated as “vesicle” only.

- In the Supporting information the authors display SAXS data at 5 oC. On the C10mix they describe the changes in integrated peak intensity at various pressure as “constant” (line 26). The figure shown displays a clear peak at 700 bar. The authors should revise their wording here and describe what is shown.

- In the Supporting information, line 30, it is obviously meant $q = 0.056 - 0.088$, not $q = 0.56$.

- The discussion of the changes in the integrated peak intensities, referred to as swollen phase area or collapsed phase area, is incomplete as it presents only data at 5 oC. Nearly all data discussed in the paper are at 20 oC or higher and it would be interesting to have full data at least at 20 oC here. In Fig S5 the FWHM of the peaks is given without further description or presentation of the corresponding SAXS curves.

- The statement that the changes depicted show a “racking from one phase to the others” remains unclear (line 39). It can also be debated if the peak integrated intensity can be directly related to the ratio of the two phases in the systems and given as area of the two phases. Also, to term this ratio as area of the phase is misleading.

- In line 51 the authors state the collapsed phase in the systems with alkanes disappear at temperatures above 20 oC. This is shown in Fig. 4 of the paper but no further experimental examples are given. Additional proof as the reference to the previous paper of the authors would be helpful.

- The discussion on the d-spacing of the collapsed phase displayed in Fig. S4 is of no meaning as there are only speculations about the differences between the C10mix system and the samples with alkanes (line 55 ff). What is meant with difference in molecular partitioning. Is there any evidence available?

- The presentation and discussion of the changes in the FWHM of the correlation peak of the swollen phase obtained at 20 oC (line 73 ff, Fig. S5) is not very distinct. It is unclear the relation between the FWHM of a peak and any intermembrane correlation as indicated with the derived d-spacing. The broadening of the correlation peak (i.e. increase in FWHM) indicates a reduction in intermembrane interaction. To relate this with any phase transition is arbitrary without further evidence as e.g. shown in Fig. 5 and discussed there. There a phase transition is considered at 20 oC and 400 bar. In Fig. S5 such a transition seems to happen at 20 oC at 700 bar. The comment in line 85 ff that it does not have any impact on the findings and interpretation of the d-spacing of the swollen phase is not convincing based on the shown inconsistencies in the interpretation of the data.

Reviewer #3 (Remarks to the Author):

The authors used small angle x-ray scattering (SAXS) and elastic incoherent neutron scattering (EINS) for investigating the effect of the presence of small amounts of selected linear or branched alkanes in multilamellar vesicle dispersion of a 1:1 molar mixture of decanoic acid and decanol in bicine buffer (0.2 M), pH 8.5. The work is a follow-up investigation of the study on the temperature stability of alkane-free decanoic acid/decanol (1:1) bilayers, published in Langmuir this year (ref. 30). This previous work and the work described in the submitted manuscript are related to the potential role of fatty acid-based membranes in prebiological times (possible formation of protocells).

The key result of the study presented in the manuscript is that the presence of the alkanes under the conditions used help stabilizing decanoic acid/decanol bilayers. This indicates that the hypothesis about the formation of early organic membranes under harsh conditions (high temperature and pressure) remains a reasonable hypothesis, despite the fact the experiments were carried out with concentrated multilamellar vesicle dispersions and not with unilamellar vesicles.

Often it is said that simple organic bilayer membranes are not stable. The data in the paper show that bilayer stability as such is not a problem. An interesting follow-up study could be to investigate the bilayer permeability in the presence of alkanes.

Although I am not a scattering expert, I got the impression that the experiments were carried out carefully and the results obtained are critically discussed. The work can be considered as motivation for those working experimentally on protocell models, possibly exploring the stability of these systems when in unilamellar, cell-mimicking state.

I have only a couple of observations/suggestions. Publication after minor revisions is recommended:

1. The last sentence of the abstract is not clear. Please improve! "Have a lower swelling" What does it mean? "... lower swelling less affected ..." as compared to what? Is reference made to an alkane-free system?
2. Page 1, line 2: There are a many more researcher working on prebiotic chemistry than Miller and Orgel. Sutherland, for example, is very active since many years. Saladino is another active researcher.
3. Page 1, line 33. Discussing about protocellular compartmentalization, Morowitz, Deamer, Luisi, and Szostak contributed significantly, including the aspect vesicle compartment reproduction. A key paper is the article by Szostak, Bartel, Luisi, published in Nature in 2001.
4. Page 2, line 55: remove ^{...}
5. Page 4, line 118: 0.2 M bicine buffer was used, adjusted to pH = 8.5. Concerning pD: Did you just use the pH-meter reading when preparing samples with deuterated water. This has to be explained/clarified. What about pH/decanoic acid pKa changes with increasing temperature (Fig. 6). Ignored? Please clarify!

Reviewers' comments:

Reviewer #1 (Remarks to the Author):

The authors show data that demonstrate the reduced mobility of a model bilayer system composed of capric acid and decanol, that is aimed to resemble the first bilayer structures formed in the process for life formation. These two molecules experience increased movement upon increasing temperature, and this is counteracted by pressure. Life is thought to have formed in regions of high temperature and pressure, therefore the authors study the effect of bilayer structure (in terms of d-spacing) and bilayer motions (general motions) upon incorporating two alkanes: a linear one and a branched one. BY using SAXS, the authors follow the change in d-spacing and observe how the alkanes decrease this value at all temperatures and pressures. By using EINS and perdeuterated alkane, the collective motion of the acyl chain in capric acid and decanol were shown to be maintained at all temperatures and pressures to values similar to those at 25C. This was in contrast with the samples measured without added alkane, that show a continuous increase in motion with temperature at all pressures. Together the data show that the presence of the alkanes reduces the bilayer mobility (and increase bilayer rigidity) enabling the bilayer structure to be retained even under very harsh conditions. The work is very convincing and well presented. I wonder what the authors think is the structure of these bilayers? Where are the added alkanes positioning? Will they be freely mixed in the bilayer structure or is there a preference to occupy a certain position within the bilayer?

We thank the reviewer for his/her time to read the manuscript and the encouraging remarks.

The positioning of squalane molecules inside the bilayer was studied on model membranes made by DOPC/DOPG (Hauß et al., *Biochim. Biophys. Acta* **2002**, 1556, 149-154) and DoPhPC:DoPhPE (Salvador et al., *Langmuir* **2020**, 36, 7375-7382). Both studies found that this alkanes has a preference to be in the mid-plane of the bilayer, perpendicular to the lipid hydrophobic chains. We could not perform a similar investigation on the localization of alkane in our proto-lipid system because of the impossibility to create hydrated multilayers made of decanoic acid:decanol on a solid substrate, which is a prerequisite in order to localize them as described in the two above mentioned publications. In addition, several studies performed in the 80s (e.g. Haydon et al., *BBA-Biomembranes* **1977**, 470(1), 17-34.) on the mechanisms of anesthesia by alkanes have pointed out that long chain alkanes did insert into the midplane of the membrane, thereby disrupting the transmission of the electric signal, although they did not localize the alkanes with precision. Thus, although referring to different membrane systems, they point towards such a preferential position.

There are still some questions I hope to be addressed prior to submission:

-state the number of samples measured.

We have added the number and type of samples studied at the end of the "Sample preparation" section as follows:

"For the complementary neutron scattering experiments, the samples were prepared at 80 mM concentration. The three model membranes measured were:

1. C10 mix
2. C10 mix + 2% (h/d) eicosane

3. C10 mix + 2% (h/d) squalane”

-state how the error bars in Figure 5 and 6 are deduced from.

The experimental section was modified accordingly as follows:

“The parameters corresponding to the center of the Gaussian functions were converted into d-spacing values (via $d_{spacing} = 2\pi/q$), as well as the associated errors (through error propagation)”

“This gives a direct way of measuring the MSD and the corresponding error by simple error propagation from I_{sum} and allows to profit from a better statistics. The errors on I_{sum} are only dependent on the Poisson distribution of the scattered neutrons.”

-why are the error bars in Figure 6 for data in absence of added alcane so much larger than for the data with added alcane?

This is mainly due to the significantly lower scattering intensity of the C10 mix sample (visible from new Figure 4). However, in order to better address the criticism of both reviewers #1 and #2 regarding the large error bars and the overall quality of the EINS results, we decided to re-analyze the data using an alternative methodology which has been proposed and published by Zeller et al., J. Chem. Phys., **2018**, 149, 234908.

The previous method is known as Gaussian approximation (GA), where the Mean Square Displacement (MSD) is calculated from the slope of the I_{inc} curves (as in Figure 3) and whose errors are the ones arising from the fits. In the new method the MSD is directly defined from the inverse of the square of I_{inc} summed over the scattering angles validated for the GA (indeed this new definition holds also in the limits of q given by the GA, see equation 19 of the given ref.).

The advantage of the new method is that the errors on the MSD are found directly from the errors in the I_{inc} (which are calculated from the Poisson distribution). However this new method is defined to within an arbitrary normalization factors. Therefore, we have re-analyzed our data and used the results found from the GA to set this normalization factor (same for all samples and p-T points). We show below the comparisons between the former and the new analysis as information for the reviewer, as well as the new plots which will substitute the ones on Figure 6 and 7 (new Figure 7 and 8) in the manuscript.

Corrected Figure 6 (now Figure 7)

Corrected Figure 7 (now Figure 8)

We modified the wording in the manuscript as below:

“Each I_{inc} vs q curve obtained per p - T point was analysed using a model-free approach described in 46. In the limits of the Gaussian approximation (GA), which assumes harmonic motions of the atoms around their equilibrium positions and thus a linear behaviour of $\log(I_{inc})$ vs. q^2 , one can write the sum of the intensity curves over q , namely $I_{sum}(T) = \sum_{q_{min}}^{q_{max}} I_{inc}(q, T)$, as follows:

$$I_{sum}^2(T) \propto \frac{1}{\langle u^2 \rangle} \text{Å}^2 \quad (2)$$

with $\langle u^2 \rangle$ the hydrogen MSD and Å the angstrom dimension. This gives a direct way of measuring the MSD and the corresponding error by simple error propagation from I_{sum} and allows to profit from a better statistics.”

“A normalization factor, common to all p-T points and samples measured, was applied on the MSD values to obtain results quantitatively comparable with the ones found from the standard GA fits (via the formula $I_{inc}(q) \cong I_0 \exp[-\langle u^2 \rangle q^2 / 3]$). The latter fits lead to absolute $\langle u^2 \rangle$ values (and thus can be used for normalization) but they are highly affected by the fitting errors, justifying our use of the alternative fit- and model-free method.”

We thank Reviewer 1 for requiring to improve our analysis, which led to better quality results than initially presented. We modified the manuscript accordingly.

-what is the purity of the perdeuterated eicosane?

We have added this information in the manuscript as below:

“The chemical and isotopic purity (CP - IP) of the used compounds are the following:

- sodium decanoate: $\geq 98\%$ CP;
- 1-decanol: $\geq 98\%$ CP;
- h-eicosane: 99% CP;
- d-eicosane: 98% CP; 98% IP;
- h-squalane: 96% CP;
- d-squalane: 98% CP; 98% IP;
- bicine: $\geq 99\%$ CP.”

-the condensed phase formed? is it thought as two different vesicles populations, one that falls out of solution and another one that remains in solution (the fluid phase)?

We observed the appearance of a collapsed phase that forms at $T = 5^\circ\text{C}$. We identified this in our former study (Misuraca et al., Langmuir, **2020**, 36, 45, 13516–13526) as an effect of the flocculation of the decanoic acid. The flocculation is the result of the crossing of the fatty acid Krafft temperature, which is the temperature below which the acid solubility is below its critical micelle concentration. Based on the observed linearity of the Krafft temperature with the acid chain length (Fameau et al., Adv. Coll. Int. Sci. **2014**, 207, 43-64), for the decanoic acid it is expected to be at $T \approx 10^\circ\text{C}$. Yes, it can be thought as two different populations: MLVs that remain suspended in solution and the flocs phase, which is not macroscopically separated, hence the milky aspect of the solution.

As both reviewers #1 and #2 expressed the need for clarification about the collapsed phase, we have added a clarifying sentence in the manuscript as follows:

“Therefore, at $T = 5^\circ\text{C}$ we are in presence of a two-phase system: one is the MLV phase, with weakly interacting bilayers that leads to the broad correlation observed at lower q (swollen phase); the other one, which can be thought as made of microscopically phase separated aggregates, that gives a sharp correlation in the scattering curve at higher q (collapsed phase).”

Reviewer #2 (Remarks to the Author):

The paper submitted by Misuraca et al. is dealing with the structure and dynamics of protomembrane model systems in order to get some insight into early strategies by nature to adopt functional membranes surviving the harsh environment of early life. The authors have applied small angle x-ray scattering (SAXS) and elastic incoherent neutron scattering (EINS) to probe structure and dynamics of simple model systems of mixed fatty acid / fatty alcohol systems in the presence of certain alkanes at various temperature and pressure values. The found effects on the structure and the local dynamics of the alky chains in these model systems indicate, argued by the authors, a stabilizing effect of the added alkanes on the membrane resistance towards temperature and pressure changes. This behavior is seen as a possible strategy of survival for the initial forms of life.

The paper is reasonable well organized and the description of the experiments conducted is quite sound. In order to consider a publication of the work presented nevertheless a number of improvements should be taken into account, as the presentation and discussion of some of the results should be improved critically. In the following these points will be discussed.

We thank the reviewer for his/her time to read the manuscript and the constructive criticism.

- In the introductory part the authors describe the reduction in intermembrane space as "a sign of its (alkanes) role in dampening membrane fluctuation". Usually large intermembrane spacings are an effect of electrostatic repulsion and are triggered mainly by changes in ionic strength or pH. The large d-spacing observed and related to a so called "swollen phase" may indicate a weak correlation between individual membranes which might be a transient interaction averaged over the time of the sampling of the SAXS experiment. The shown changes in d-spacing (Fig. 5) may reflect this transient correlation and the found decrease by presence of alkanes could be interpreted as a damping of membrane fluctuations or more general of the dynamic of the membrane fluctuations.

On the other hand the temperature dependence of this interaction seems to get lost in the presence of the alkanes as on changing the temperature from 20 to 35 °C has no substantial effect anymore on the d-spacing (Fig.5, middle). It is not clear why this should happen and the simple assumption of the damping of membrane fluctuations seems to be more complex. There is also no substantial effect in the MSD shown in Fig. 6 on increasing the temperature in the systems with eicosane, which should exhibit an increase. Here, the authors should revise their interpretations and be more careful in the description of the drawn conclusion on this behavior.

Effectively the pH and ionic strength modulate the electrostatic interactions in charged systems, pH being responsible for the net lipid charge and the ionic strength for its screening by counter-ions. However, it is worth noticing that our systems are prepared in a rather high concentration buffer (0.2M bicine), therefore electrostatic interactions between subsequent membranes should be partly if not entirely screened. Moreover, it has been shown by Kapoor et al. (Angew. Chemie Int. Ed. **2014**, 53, 8397-8401), that unlike the vesicles made by decanoic acid that are heavily dependent on the solution pH, the ones consisting of the mixture decanoic acid:decanol 1:1 (which we call C10 mix) are stable in a wide pH range (6 – 12). We have chosen pH (pD) 8.5 to be in a safe region where minor changes in the buffer pK_a (dpK_a/dT = -0.018, Good et al., Biochemistry, **1966**, 5(2), 467-477) and the solution pH given by the

temperature increase do not affect the vesicle stability. Therefore, as the reviewer states, the changes in d-spacing shown in Figure 5 (new Figure 6) when the alkanes have been incorporated can be interpreted as a dampening of the membrane fluctuations.

Although small, the temperature effect is still visible in the general trend of the data at 20 and 35 °C in the sample with eicosane (new Figure 6). In fact, such difference would have been similarly small (but significant) in the C10 mix sample between 20 and 35 °C if the sample at 20 °C did not undergo the phase transition. Thus, the increase in the average d-spacing between 20 and 35 °C (left and middle) and also from 35 to 50 ° (all the three samples) is of comparable extent on the different samples.

About the EINS data on Figure 6 and 7 (new Figures 7 and 8), to better address the criticism of both reviewers #1 and #2 regarding the large error bars and the overall quality of the EINS results, we decided to re-analyze the data using an alternative methodology which has been proposed and published by Zeller et al., *J. Chem. Phys.*, **2018**, 149, 234908.

The previous method is known as Gaussian approximation (GA), where the Mean Square Displacement (MSD) is calculated from the slope of the I_{inc} curves (as in Figure 3) and whose errors are the ones arising from the fits. In the new method the MSD is directly defined from the inverse of the square of I_{inc} summed over the scattering angles validated for the GA (indeed this new definition holds also in the limits of q given by the GA, see equation 19 of the given ref.).

The advantage of the new method is that the errors on the MSD are found directly from the errors in the I_{inc} (which are calculated from the Poisson distribution). However this new method is defined to within an arbitrary normalization factors. Therefore, we have re-analyzed our data and used the results found from the GA to set this normalization factor (same for all samples and p-T points). We show below the comparisons between the former and the new analysis as information for the reviewer, as well as the new plots which will substitute the ones on Figure 6 and 7 (new Figure 7 and 8) in the manuscript.

New Figure 6

New Figure 7

We modified the wording in the manuscript as follows:

“Each I_{inc} vs q curve obtained per p - T point was analysed using a model-free approach described in 46. In the limits of the Gaussian approximation (GA), which assumes harmonic motions of the atoms around their equilibrium positions and thus a linear behaviour of $\log(I_{inc})$ vs. q^2 , one can write the sum of the intensity curves over q , namely $I_{sum}(T) = \sum_{q_{min}}^{q_{max}} I_{inc}(q, T)$, as follows:

$$I_{sum}^2(T) \propto \frac{1}{\langle u^2 \rangle} \text{Å}^2 \quad (2)$$

with $\langle u^2 \rangle$ the hydrogen MSD and Å the angstrom dimension. This gives a direct way of measuring the MSD and the corresponding error by simple error propagation from I_{sum} and allows to profit from a better statistics.”

“A normalization factor, common to all p-T points and samples measured, was applied on the MSD values to obtain results quantitatively comparable with the ones found from the standard GA fits (via the formula $I_{inc}(q) \cong I_0 \exp[-\langle u^2 \rangle q^2 / 3]$). The latter fits lead to absolute $\langle u^2 \rangle$ values (and thus can be used for normalization) but they are highly affected by the fitting errors, justifying our use of the alternative fit- and model-free method.”

We thank Reviewer 2 for requiring to improve our analysis, which led to better quality results than initially presented. We modified the manuscript accordingly.

As a concluding remark (now better visible with the new analysis) the sample with eicosane still shows an increase in the MSD with temperature increase (the derivative $d \text{MSD} / dT$ is in fact always positive), but of a much lower extent.

- The authors should make a comment regarding beam damage onto the samples by the exposure to the synchrotron radiation.

The I22 beamline at Diamond Light Source (Didcot, UK) is equipped with a fast shutter that is used to make sure that the samples are only illuminated during the short counting times and not during T-p changes and equilibration time. The 17 keV x-rays used for the high pressure experiments on I22 allow reduce the radiation damage to biological samples (Brooks et al., Rev. Scien. Inst., **2010**, 81(6), 064103.). If radiation damage occurred to the samples, this should be visible in the SAXS curves as a decrease in the scattering intensity. Considering that the scanning was done as isotherms from $T = 5 \text{ °C}$, the Figure below compares the scattering intensity at $T = 5 \text{ °C}$ and $p = 10 \text{ bar}$ with the one at $T = 50 \text{ °C}$ and $p = 1000 \text{ bar}$ (last useful scan in the analysis that leads to Figure 5 (new Figure 6) data in the manuscript). No decrease in intensity was observed. We clarified this now in the main text, adding the following paragraph:

“The I22 beamline at Diamond Light Source (Didcot, UK) is equipped with a fast shutter that is used to make sure that the samples are only is illuminated during the short counting times and not during T-p changes and equilibration time. The 17 keV x-rays used for the high pressure experiments on I22 helps reduce the radiation damage to biological samples. If radiation damage occurred to the samples, this would have resulted in a decrease of the SAXS intensity versus time since the T-p scans where performed on the same sample for a given composition. No decrease in intensity was observed.”

- The authors have not tried to do a full fitting of the collected SAXS curves (see p.4, line 153). With relation to the assumed changes in the interaction between the individual bilayers in the MLV it would be interesting if the sizes of the MLV can be fitted with a classical sphere or ellipsoidal model, which would indicate a shape fluctuation. It would also be interesting to have seen some electron microscopy pictures to prove the multilamellar arrangement proposed.

It is not possible to fit the scattering curves with a sphere or ellipsoidal model, because the size is convoluted by the intrinsic high polydispersity of the MLVs, therefore the information on size and shape fluctuations is not accessible. A full fit of the small angle scattering curve can only be done in case of monodisperse or moderately polydisperse samples, as it was done for instance in our previous study (see attachment Misuraca et al., *Langmuir*, **2020**, 36, 45, 13516–13526) on samples that were extruded for this purpose. Also in that article it is possible to see (Figure 6 of the *Langmuir* paper, right panel) that the C10 mix at $T \geq 20$ °C has two orders of correlation, with the 2nd order at $q_{2nd} = 2 * q_{1st}$. Therefore, there is no doubt that we are in presence of MLVs.

To further answer the reviewer's question (since it cannot be assumed that the alkanes do not trigger any non-lamellar phase formation) we enclose new SANS data of C10 mix, C10 mix + 2% eicosane and C10 mix + 2% squalane at 80 mM concentration collected at $T = 20$ °C. As visible from the Figure below (in $I * q^2$ representation), all samples show two orders of the MLVs' membrane correlations and the position of the 2nd order at $q_{2nd} = 2 * q_{1st}$ ensures that all phases are lamellar. We included this additional data in the manuscript (new Figure 2), and adding the following sentence:

“We verified that the observed correlation is due to a lamellar phase of MLVs from the position of the 2nd order of the correlation with respect to the 1st order: $q_{2nd} = 2 * q_{1st}$ (Figure 2).”

- On p.7, line 222 the authors should write “a drop of line towards zero” as zero is not reached.

We added this correction: “The $q^2 > 2 \text{ \AA}^{-2}$ data for the C10 mix at $T = 84 \text{ }^\circ\text{C}$ approach zero and are not visible in the $\log(I_{inc})$ representation.”

- The description of the collapsed phase as a flocculated fraction of the systems is difficult to follow (p.7, line 236 ff). Do the authors mean a two phase system or a phase separation? This should be explained more clearly.

We observed the appearance of a collapsed phase that forms at $T = 5 \text{ }^\circ\text{C}$. We identified this in our former study (Misuraca et al., *Langmuir*, **2020**, 36, 45, 13516–13526) as an effect of the flocculation of the decanoic acid. The flocculation is the result of the crossing of the fatty acid Krafft temperature, which is the temperature below which the acid solubility is below its critical micelle concentration. Based on the observed linearity of the Krafft temperature with the acid chain length (Fameau et al., *Adv. Coll. Int. Sci.* **2014**, 207, 43-64), for the decanoic acid it is expected to be at $T \approx 10 \text{ }^\circ\text{C}$. Therefore, at $T = 5 \text{ }^\circ\text{C}$ we observe a phase separation which leads to two phases: MLVs that remain suspended in solution and the flocs phase, which is not macroscopically separated, hence the milky aspect of the solution.

As both reviewers #1 and #2 expressed the need for clarification about the collapsed phase, we have added a clarifying sentence in the manuscript as follows:

“Therefore, at $T = 5\text{ }^{\circ}\text{C}$ we are in presence of a two-phase system: one is the MLV phase, with weakly interacting bilayers that leads to the broad correlation observed at lower q (swollen phase); the other one, which can be thought as made of microscopically phase separated aggregates, that gives a sharp correlation in the scattering curve at higher q (collapsed phase).”

- The authors mention a weak undulation in the q range $0.1\text{-}0.2\text{ Ang-}^{-1}$ (p.8, line 248). This is in no way visible in the presented graphs. The authors should give more clear evidence on this assumption.

We have modified the Figure 4 (new Figure 5) to allow better visual inspection. We added a more exhaustive description of the features and the trends that are visible and at which temperatures, as below:

“The curve at $T = 20\text{ }^{\circ}\text{C}$ (Figure 5) captures an intermediate state where the melting of the collapsed phase is almost completed. At $T = 35\text{ }^{\circ}\text{C}$, one main correlation is observed at $q \approx 0.08\text{ \AA}^{-1}$ (the 2nd order can be detected at $q \approx 0.16\text{ \AA}^{-1}$ although weak. At $T = 50\text{ }^{\circ}\text{C}$, the correlation is further shifted to lower q and broadened, hardly detectable by visual inspection. At $T = 65\text{ }^{\circ}\text{C}$ one can only guess the position from the trend at low temperature, and finally at $T = 80\text{ }^{\circ}\text{C}$ the correlation is completely lost. This behaviour, with the position of the correlation shifting to lower q until disappearing, is expected for MLVs that undergo swelling upon temperature increasing, until unbinding. In our analysis, we calculated the d -spacing of the three samples at $T \leq 50\text{ }^{\circ}\text{C}$ only.”

- On p.8, line 250 ff the q and d -spacing numbers are missing in the text. The values are now added:

“The data from the C10 mix sample at $T = 20\text{ }^{\circ}\text{C}$ and $p = 10\text{ bar}$ show a swollen, broad phase centred at $q \approx 0.05\text{ \AA}^{-1}$ (d -spacing $\approx 122\text{ \AA}$) together with a small, thin correlation at $q \approx 0.08\text{ \AA}^{-1}$ (d -spacing $\approx 75\text{ \AA}$).”

- The authors state the fluid-gel transition at $35\text{ }^{\circ}\text{C}$ to be at 1000 bar . Following Fig. 5 this transition is at about 500 bar . In Fig. 5 the authors should not connect the points measured by arbitrary curves but with straight lines to visualize the phase changes and error ranges.

We recognize that a precise definition of the transition temperature is difficult from our data. Therefore, we have modified the manuscript by making qualitative considerations and clarifying the expected shift in the temperature range of the transition. We also modified the Figure 5 (new Figure 6) accordingly. Moreover, we add new FTIR data on the C10 mix sample in the Supp. Info. (Fig. S6) that also show the occurrence (and the estimate temperature) of the fluid-gel transition.

The manuscript has been modified as follows:

“Here the transition seems to happen at $p \approx 300\text{ bar}$ at $T = 20\text{ }^{\circ}\text{C}$ in agreement with FTIR measurements (Supp. Info.). Assuming the linearity of the fluid-gel phase transition with p - T variation, this leads to a shift of $\approx 3\text{ }^{\circ}\text{C} / 100\text{ bar}$. This value is similar with what has been observed with phospholipid membranes

($\approx 2 \text{ }^\circ\text{C} / 100 \text{ bar}$). Following this relationship, the phase transition of the C10 mix at $T = 35 \text{ }^\circ\text{C}$ is expected at $p \approx 800 \text{ bar}$, although our data do not allow to conclude unambiguously.”

- The results of the EINS data shown in Fig. 6 imply a linear change in MSD for the mixed system without alkanes for different pressure applied. A more closer look at the data shows an increase between 25 to 45 oC and then a roughly linear change or even decrease within the error range given towards higher temperatures. It looks as if the simple conclusion of a linear increase in MSD is not fully valid. The authors should carefully reanalyze this data as it seems to resample partly the different behavior in d-spacing with temperature shown in the SAXS data in Fig.4 and changed due to the fluid-gel transition at applied pressure.

As suggested by the reviewers #1 and #2, we have reanalyzed the data as described in detail in a previous answer. The quality of the newly analyzed results allow us to better define the trend of the MSD for each curve. The reviewers are gratefully acknowledged to push us to do that.

Regarding the conclusive remark, please note that the Figure 5 (new Figure 6) and Figure 6 (new Figure 7) are plotted as function of p and T respectively, therefore are not observing the same points in p - T . The transition observed in Figure 5 (new Figure 6) at $p \approx 300 \text{ bar}$ and $T = 20 \text{ }^\circ\text{C}$, knowing that the effects of pressure increase are relatable to the effects of decreasing temperature, is expected at $T < 20 \text{ }^\circ\text{C}$, a region not explored from Figure 6 (new Figure 7) data.

- On page 11, line 327 ff the authors claim squalene has the biggest effect on the structure and dynamics investigated. Unfortunately, the authors have not shown any results on the EINS with squalene. It is assumed squalene will have a more clear or similar effect as found with eicosane, but this is only an assumption derived on the basis of the SAXS data. The authors should be slightly more cautious on this conclusion.

We corrected the manuscript to clarify that the biggest effect of the squalane refers only to the structural study done with SAXS. Below the correction:

“Squalane leads to the biggest effect on the structural MLV characterization”

- A minor point is the use of the term vesicle. To avoid confusion and to describe the work presented correctly the authors should always use the term “multilamellar vesicles” in the manuscript as only multilamellar systems have been investigated. At increased temperature the multilamellar vesicles obviously transform into unilamellar vesicles but there is no experimental proof given throughout the manuscript on this assumption, therefore it should be avoided to describe the systems investigated as “vesicle” only.

We have modified the wording whenever it referred to the samples we measured, to make sure it is clear that we used MLVs. However, we maintained the word “vesicle” whenever it referred to literature work or when talking about the form factor $P(q)$. In fact, both ULVs and MLVs are indistinguishable in terms of the form factor $P(q)$, and the difference is in the fact that the MLVs have a $S(q) \neq 1$.

- In the Supporting information the authors display SAXS data at 5 oC. On the C10mix they describe the changes in integrated peak intensity at various pressure as “constant” (line 26). The figure shown

displays a clear peak at 700 bar. The authors should revise their wording here and describe what is shown.

The error bar shown in the plot of Figure S1 was to give an idea of the uncertainty of each point of the “collapsed phase area” of the C10 mix sample. Therefore what is observed at 700 bar is inside the uncertainty in the determination of the area. We clarified this aspect by adding now all error bars to avoid misleading the reader.

- In the Supporting information, line 30, it is obviously meant $q = 0.056 - 0.088$, not $q = 0.56$.

We thank the reviewer for the remark and corrected the text.

- The discussion of the changes in the integrated peak intensities, referred to as swollen phase area or collapsed phase area, is incomplete as it presents only data at 5 oC. Nearly all data discussed in the paper are at 20 oC or higher and it would be interesting to have full data at least at 20 oC here. In Fig S5 the FWHM of the peaks is given without further description or presentation of the corresponding SAXS curves.

The discussion about the collapsed and swollen phases refers only to the data at $T = 5\text{ }^{\circ}\text{C}$, because it is not observed at all the other temperatures on eicosane and squalane containing samples. As proposed by the reviewer, we have included the full dataset at 20 °C for the three samples in the Supp. Info. (see new Figure S5).

Regarding Figure S5, it corresponds to the FWHM obtained by the same analysis that led to the results presented in new Figure 6 (where the q_{center} of each Gaussian was converted to d-spacing values). Following the remark of the reviewer regarding new Figure 6, we removed the trend lines also from Figure S5.

- The statement that the changes depicted show a “racking from one phase to the others” remains unclear (line 39). It can also be debated if the peak integrated intensity can be directly related to the ratio of the two phases in the systems and given as area of the two phases. Also, to term this ratio as area of the phase is misleading.

We substituted the word “racking” with “transfer”.

The fact that the d-spacing does not change, as a function of the pressure, allows us to consider the integrated intensity proportional to the phase volume fraction (otherwise this would not have been possible, as noted by the reviewer). Furthermore, the increase/decrease of the integrated intensity (and hence the corresponding phase) is to be evaluated qualitatively. We modified the manuscript to avoid confusion, as below:

“The fact that the peak of the collapsed phase is observed at a constant q -value allows us to link qualitatively the intensity of the correlation with the volume fraction of sample in that phase.”

- In line 51 the authors state the collapsed phase in the systems with alkanes disappear at temperatures above 20 oC. This is shown in Fig. 4 of the paper but no further experimental examples are given. Additional proof as the reference to the previous paper of the authors would be helpful.

We clarified this point by adding the full datasets of the 3 samples at T = 20 °C, where the disappearance of the collapsed phase is observed (new Figure S5).

- The discussion on the d-spacing of the collapsed phase displayed in Fig. S4 is of no meaning as there are only speculations about the differences between the C10mix system and the samples with alkanes (line 55 ff). What is meant with difference in molecular partitioning. Is there any evidence available?

We proposed an interpretation that would explain the significant difference between what is observed on the alkane-including samples and the C10 mix sample (i.e. the difference in d-spacing and the fact that the phase of eicosane/squalane samples disappears at $T \geq 20$ °C while it does not on the C10 mix). However, since we cannot provide additional evidence to support this interpretation, we removed the sentence and the related Figure S4.

- The presentation and discussion of the changes in the FWHM of the correlation peak of the swollen phase obtained at 20 oC (line 73 ff, Fig. S5) is not very distinct. It is unclear the relation between the FWHM of a peak and any intermembrane correlation as indicated with the derived d-spacing. The broadening of the correlation peak (i.e. increase in FWHM) indicates a reduction in intermembrane interaction. To relate this with any phase transition is arbitrary without further evidence as e.g. shown in Fig. 5 and discussed there. There a phase transition is considered at 20 oC and 400 bar. In Fig. S5 such a transition seems to happen at 20 oC at 700 bar. The comment in line 85 ff that it does not have any impact on the findings and interpretation of the d-spacing of the swollen phase is not convincing based on the shown inconsistencies in the interpretation of the data.

As stated previously, we modified Figure S5 and removed the sentences in the Supp. Info. which related the phase transition to the FWHM. By the sentence in line 85 we meant that the considerations done on the FWHM data have no impact *on the fits*. To avoiding misleading, we removed this sentence.

Reviewer #3 (Remarks to the Author):

The authors used small angle x-ray scattering (SAXS) and elastic incoherent neutron scattering (EINS) for investigating the effect of the presence of small amounts of selected linear or branched alkanes in multilamellar vesicle dispersion of a 1:1 molar mixture of decanoic acid and decanol in bicine buffer (0.2 M), pH 8.5. The work is a follow-up investigation of the study on the temperature stability of alkane-free decanoic acid/decanol (1:1) bilayers, published in Langmuir this year (ref. 30). This previous work and the work described in the submitted manuscript are related to the potential role of fatty acid-based membranes in prebiological times (possible formation of protocells).

The key result of the study presented in the manuscript is that the presence of the alkanes under the conditions used help stabilizing decanoic acid/decanol bilayers. This indicates that the hypothesis about the formation of early organic membranes under harsh conditions (high temperature and pressure) remains a reasonable hypothesis, despite the fact the experiments were carried out with concentrated

multilamellar vesicle dispersions and not with unilamellar vesicles.

Often it is said that simple organic bilayer membranes are not stable. The data in the paper show that bilayer stability as such is not a problem. An interesting follow-up study could be to investigate the bilayer permeability in the presence of alkanes.

Although I am not a scattering expert, I got the impression that the experiments were carried out carefully and the results obtained are critically discussed. The work can be considered as motivation for those working experimentally on protocell models, possibly exploring the stability of these systems when in unilamellar, cell-mimicking state.

We thank the reviewer for his/her time to read the manuscript and the encouraging remarks.

I have only a couple of observations/suggestions. Publication after minor revisions is recommended:

1. The last sentence of the abstract is not clear. Please improve! "Have a lower swelling" What does it mean? "... lower swelling less affected ..." as compared to what? Is reference made to an alkane-free system?

We rephrased the abstract to clarify, as follows: "Our data show that samples containing alkanes in the membrane have a lower multilamellar vesicle swelling induced by the temperature increase and are significantly less affected by pressure variation as compared to alkane-free samples"

2. Page 1, line 2: There are a many more researcher working on prebiotic chemistry than Miller and Orgel. Sutherland, for example, is very active since many years. Saladino is another active researcher.

We recognize that the cited references are by no means exhaustive. We have added some highly relevant studies from Sutherland and Saladino.

3. Page 1, line 33. Discussing about protocellular compartmentalization, Morowitz, Deamer, Luisi, and Szostak contributed significantly, including the aspect vesicle compartment reproduction. A key paper is the article by Szostak, Bartel, Luisi, published in Nature in 2001.

We have added this reference.

4. Page 2, line 55: remove "

It is done.

5. Page 4, line 118: 0.2 M bicine buffer was used, adjusted to pH = 8.5. Concerning pD: Did you just use the pH-meter reading when preparing samples with deuterated water. This has to be explained/clarified. What about pH/decanoic acid pKa changes with increasing temperature (Fig. 6). Ignored? Please clarify!

When adjusting the pD of the D₂O solutions to 8.5, we used the formula (Krężel et al., J. Inor. Bio., 2004, 98(1), 161-166.):

$$pD = pH^* + 0.4$$

where pH^* is the value measured by an H_2O -calibrated pH-meter.

Previous work on decanoic acid:decanol systems (Kapoor et al., *Angew. Chemie Int. Ed.* **2014**, 53, 8397-8401) have shown that, unlike the vesicles made by decanoic acid that are heavily dependent on the solution pH, the ones consisting of the decanoic acid:decanol 1:1 mixture (which we called C10 mix) are stable in a wide pH range (6 – 12). Furthermore, the temperature-dependent change in the pK_a of bicine buffer is rather low ($dpK_a/dT = -0.018$) (Good et al., *Biochemistry*, **1966**, 5(2), 467-477.).

We have chosen pH (or pD) 8.5 so to be in a safe region where minor changes in the buffer pK_a and solution pH given by the temperature increase do not affect the vesicle stability. We give now more details about it in the text, as below:

“When adjusting the pD of the D_2O solutions to 8.5, we used the formula $pD = pH^* + 0.4$, where pH^* is the value measured by an H_2O -calibrated pH-meter.

Previous work on decanoic acid:decanol systems has shown that, unlike the vesicles made by decanoic acid that are heavily dependent on the solution pH, the ones consisting of the decanoic acid:decanol 1:1 mixture (which we named C10 mix) are stable in a wide pH range (pH 6 – 12). Furthermore, the temperature-dependent change in the pK_a of bicine buffer is rather low ($dpK_a/dT = -0.018$). We have chosen pH (or pD) 8.5 so that the minor changes in the buffer pK_a and solution pH given by the temperature increase do not affect the vesicle stability.”

Reviewers' comments:

Reviewer #1 (Remarks to the Author):

The authors have exhaustively taken into accounts the comments made by the referees. When referring to the number of samples, I mean the number of repeats. I imagine this was 1 as typical for scattering experiments. This should be clarified.

Reviewer #2 (Remarks to the Author):

The authors have done reasonable work to improve the present manuscript taken into account comments and suggestions by the reviewers given. This has led to a more clear presentation and discussion of the results and clarified the content. Unfortunately some question and results are still given in the text which need further explanation or discussion. Hence, the paper will need some additional revision before it can be put forward for publication in Communication Chemistry.

The following points still need discussion and corrections:

- In Fig. 2 the authors show SANS data of the 3 systems investigated at $T = 21$ oC. There is no relevance of this data to all the other structural data presented in the manuscript and the Supporting Information, where only SAXS data are presented and discussed. As the relevance of the SANS measurements is given nowhere, the authors may either relate this data to the general discussion here or skip it from the manuscript as not relevant. The data is introduced as verification of the observed lamellar structure of MLVs, but this is also shown in the corresponding SAXS data, see Fig. 3 and Supp. Info.
- The background correction of the SAXS data to evaluate location of the correlation maxima is using a linear function (line 202, page 6). It should be mentioned here that a linear background was subtracted.
- The authors discuss data of the C10 mix sample at $T = 20$ oC and $p = 10$ bar without presenting the data nowhere in the manuscript. Either this is an error and the data refer to $p = 100$ bar as shown in Fig. S4 left, or the data is missing and has to be added.
- In Fig. S4 legends to the pressure value at the curves are missing as shown in Fig. S2 and S3. The question mark in Fig. S4 should be explained in the figure legend.

Some minor points to modify:

- line 72, page 2. What is meant with the phrase "... which maxima are the conditions ..."?
- line 125, page 3. Write: "The D2O buffers allowed ...", not "This allowed ..."
- line 171, page 5. Skip "is" in "... samples are only illuminated ..."
- line 251, page 8. What is meant with "... the angstrom dimension"?
- line 254. Page 8. Add "dynamics" at "... measure of the hydrogen dynamics, ..."
- Supp. Information, line 33. Here a "shrunk phase" of the system is mentioned. Obviously the "collapsed phase" is meant as this term is used elsewhere in the paragraph. To stay consistent the term "shrunk" should be replaced.
- Supp. Information, line 100. Correct "... Figure S8 ..." to "Figure S6"

Reviewer #3 (Remarks to the Author):

Following the many suggestions made by the three reviewers, the authors made a substantial revision of the manuscript, including new analysis of the experimental data and inclusion of new figures as well as additional references to other groups working on the topics the authors refer to. The changes made are carefully explained in the rebuttal letter. It is somewhat surprising that a change in the way the data are analyzed can lead to such drastic changes in the data representation, as seen in the new Figure 8 if compared with the old Figure 7. The overall conclusions remain the same as in the originally submitted version of the manuscript. I suggest publishing the revised version of the manuscript if a SAXS expert agrees as well.

Reviewers' comments:

Reviewer #1 (Remarks to the Author):

We thank again the reviewer for his/her time to read again the manuscript and the positive feedback.

The authors have exhaustively taken into accounts the comments made by the referees. When referring to the number of samples, I mean the number of repeats. I imagine this was 1 as typical for scattering experiments. This should be clarified.

Yes, this is correct. We have clarified it in the manuscript.

Reviewer #2 (Remarks to the Author):

The authors have done reasonable work to improve the present manuscript taken into account comments and suggestions by the reviewers given. This has led to a more clear presentation and discussion of the results and clarified the content. Unfortunately some question and results are still given in the text which need further explanation or discussion. Hence, the paper will need some additional revision before it can be put forward for publication in Communication Chemistry.

We thank again the reviewer for his/her time to read again the manuscript and the constructive feedback.

The following points still need discussion and corrections:

- In Fig. 2 the authors show SANS data of the 3 systems investigated at $T = 21\text{ }^{\circ}\text{C}$. There is no relevance of this data to all the other structural data presented in the manuscript and the Supporting Information, where only SAXS data are presented and discussed. As the relevance of the SANS measurements is given nowhere, the authors may either relate this data to the general discussion here or skip it from the manuscript as not relevant. The data is introduced as verification of the observed lamellar structure of MLVs, but this is also shown in the corresponding SAXS data, see Fig. 3 and Supp. Info.

The SANS data shown in Figure 2 demonstrate that the system arranges into MLVs at ambient temperature and pressure. In Figure 3 the 2nd order of the correlation is clearly observed only on the data at $p = 1000\text{ bar}$, while its position can only be guessed at other pressure values. This is why we have added the SANS data of Figure 2, to show that the lamellar arrangement is present also at ambient temperature and pressure. We have clarified this with the following sentence:

“We verified that the observed correlation is due to a lamellar phase of MLVs from the position of the 2nd order correlation peak with respect to the 1st order: $q_{2\text{nd}} = 2 * q_{1\text{st}}$ at ambient pressure (Figure 2) as well as at high pressure (Figure 3).”

- The background correction of the SAXS data to evaluate location of the correlation maxima is using a linear function (line 202, page 6). It should be mentioned here that a linear background was subtracted.

We clarified that the q^{-k} decay is observed as a linear decay in the log-log representation, by adding the following sentence:

“Such q^{-k} trend is observed as a linear background in log-log plots.”

- The authors discuss data of the C10 mix sample at $T = 20\text{ °C}$ and $p = 10\text{ bar}$ without presenting the data nowhere in the manuscript. Either this is an error and the data refer to $p = 100\text{ bar}$ as shown in Fig. S4 left, or the data is missing and has to be added.

We refer indeed to the data shown in Figure S4. We added this reference in the main text.

- In Fig. S4 legends to the pressure value at the curves are missing as shown in Fig. S2 and S3. The question mark in Fig. S4 should be explained in the figure legend.

We added the legend and the explanation for the question mark.

Some minor points to modify:

- line 72, page 2. What is meant with the phrase "... which maxima are the conditions ..."?

We modified the sentence to clarify, as follows:

“The maximum values of both variables are the conditions expected in the proximity of the hot vents.”

- line 125, page 3. Write: "The D2O buffers allowed ...", not "This allowed ..."

We made this correction.

- line 171, page 5. Skip "is" in "... samples are only illuminated ..."

We made this correction.

- line 251, page 8. What is meant with "... the angstrom dimension"?

We modified "dimension" with "unit"

- line 254. Page 8. Add "dynamics" at "... measure of the hydrogen dynamics, ..."

We made this correction.

- Supp. Information, line 33. Here a "shrunken phase" of the system is mentioned. Obviously the "collapsed phase" is meant as this term is used elsewhere in the paragraph. To stay consistent the term "shrunken" should be replaced.

We made this correction.

- Supp. Information, line 100. Correct "... Figure S8 ..." to "Figure S6"

We made this correction.

Reviewer #3 (Remarks to the Author):

Following the many suggestions made by the three reviewers, the authors made a substantial revision of the manuscript, including new analysis of the experimental data and inclusion of new figures as well as additional references to other groups working on the topics the authors refer to. The changes made are carefully explained in the rebuttal letter. It is somewhat surprising that a change in the way the data are analyzed can lead to such drastic changes in the data representation, as seen in the new Figure 8 if compared with the old Figure 7. The overall conclusions remain the same as in the originally submitted version of the manuscript.

I suggest publishing the revised version of the manuscript if a SAXS expert agrees as well.

We thank again the reviewer for his/her time to read again the manuscript and the positive feedback.

REVIEWERS' COMMENTS:

Reviewer #2 (Remarks to the Author):

The revised manuscript by the authors have taken into account the comments and suggestions by the reviewers given. This has clarified some open points and descriptions raised by the reviewers and the paper can be considered for publication in Communication Chemistry. Nevertheless some minor revisions should still be taken into account and the authors should carefully discuss the suggested changes before final publication of the manuscript as given below.

- The sentence starting line 73 "The assemblies used were ..." is slightly confusing by logic and should be modified as follows:

The assemblies used were decanoic acid:decanol mixtures (1:1 mol/mol), similar to the ones studied by Kapoor et al. 33, hereafter called C10 mix, which appear to be the most promising in terms of vesicle stability 34, in presence of the apolar molecule eicosane, the linear 20-Carbon alkane (2 mol % of C10 mix), or squalane, the same-length branched 30-Carbon alkane following the hypotheses of Cario 28.

- The sentence starting line 86 "Furthermore, the dynamical ..." till end of the next sentence line 91 should be modified as follows:

Furthermore, the results of the dynamical study are in line with what we found for the structure modifications of the C10 mix with and without the eicosane. The average hydrogen dynamics of the membrane tails is significantly affected by temperature and pressure only when the alkane (eicosane) is not included in the sample.

- Line 184 should be modified as follows: "... inter-membrane structure factor which is of particular interest ..."

- In the legend in Fig. 3 the red lines should be named as fits to the data. Unfortunately, this is only visible in colored printing and is lost in b/w printing.

- Line 297 the bracket is missing "... 94 Å, respectively)34, and ..."

- In the Supporting Information on page 2, third paragraph the text refers to SAXS curves at $p = 1$ bar in Fig. S2 and S3. In these figures no SAXS curve at 1 bar is shown. Either the authors mean the 10 bar SAXS curves shown or the missing curves should be added to the figures.

- In the figure legend of S4, Supporting information it should be read in the last sentence: "The data of C10 mix + 2% h-squalene were not used in the analysis (as specified in the main text), because a single ..."

- Supporting Information page 3, fourth last line it should read: "... coexistence at $T = 35$ oC, what is easily visible ..."

- Supporting Information page 6, second line a blank is missing "... Alabaster, AL) to give ..."

- The authors claim in the Supporting Information page 6 that a phase transition is occurring between 200 and 400 bar. Following Fig S6, right this transition seems to happen between 300 and 700 bar. The authors should comment or modify this statement.

Reviewers' comments:

Reviewer #2 (Remarks to the Author):

We thank again the reviewer for his/her time to read again the manuscript and the positive feedback.

The revised manuscript by the authors have taken into account the comments and suggestions by the reviewers given. This has clarified some open points and descriptions raised by the reviewers and the paper can be considered for publication in Communication Chemistry. Nevertheless some minor revisions should still be taken into account and the authors should carefully discuss the suggested changes before final publication of the manuscript as given below.

- The sentence starting line 73 "The assemblies used were ..." is slightly confusing by logic and should be modified as follows:

The assemblies used were decanoic acid:decanol mixtures (1:1 mol/mol), similar to the ones studied by Kapoor et al. 33, hereafter called C10 mix, which appear to be the most promising in terms of vesicle stability 34, in presence of the apolar molecule eicosane, the linear 20-Carbon alkane (2 mol % of C10 mix), or squalane, the same-length branched 30-Carbon alkane following the hypotheses of Cario 28.

We modified the text as recommended, with a small variation as below:

"The assemblies used were decanoic acid:decanol mixtures (1:1 mol/mol), similar to the ones studied by Kapoor et al.33, hereafter called C10 mix, which appear to be the most promising in terms of vesicle stability34, in presence of the apolar molecule eicosane, the linear 20-Carbon alkane (2 mol % of C10 mix), or squalane, *similar* length branched 30-Carbon alkane following the hypotheses of Cario28"

- The sentence starting line 86 "Furthermore, the dynamical ..." till end of the next sentence line 91 should be modified as follows:

Furthermore, the results of the dynamical study are in line with what we found for the structure modifications of the C10 mix with and without the eicosane. The average hydrogen dynamics of the membrane tails is significantly affected by temperature and pressure only when the alkane (eicosane) is not included in the sample.

We modified the text as recommended.

- Line 184 should be modified as follows: "... inter-membrane structure factor which is of particular interest ..."

We modified the text as recommended.

- In the legend in Fig. 3 the red lines should be named as fits to the data. Unfortunately, this is only visible in colored printing and is lost in b/w printing.

We added the remark as recommended. We modified the color of the fitting curves from red white to maximize the contrast so that they are visible also in b/w printing.

- Line 297 the bracket is missing "... 94 Å, respectively)³⁴, and ..."

We made the correction.

- In the Supporting Information on page 2, third paragraph the text refers to SAXS curves at $p = 1$ bar in Fig. S2 and S3. In these figures no SAXS curve at 1 bar is shown. Either the authors mean the 10 bar SAXS curves shown or the missing curves should be added to the figures.

We meant $p = 10$ bar, we corrected the text.

- In the figure legend of S4, Supporting information it should be read in the last sentence: "The data of C10 mix + 2% h-squalene were not used in the analysis (as specified in the main text), because a single ..."

We modified the text as recommended.

- Supporting Information page 3, fourth last line it should read: "... coexistence at $T = 35$ °C, what is easily visible ..."

We modified the text as recommended.

- Supporting Information page 6, second line a blank is missing "... Alabaster, AL) to give ..."

We made the correction.

- The authors claim in the Supporting Information page 6 that a phase transition is occurring between 200 and 400 bar. Following Fig S6, right this transition seems to happen between 300 and 700 bar. The authors should comment or modify this statement.

We substituted the term "occurring" with "starting", to highlight that we refer to where the onset of the phase change appears and not where its mid-point is located.